# Generalized Polyspike Pattern in EEG Due to Aseptic Meningoencephalitis

**DOI:** 10.3390/diagnostics13152569

**Published:** 2023-08-02

**Authors:** Markus A. Hobert, Justina Dargvainiene, Nils G. Margraf

**Affiliations:** 1Department of Neurology, Christian-Albrecht University of Kiel and University Medical Center Schleswig-Holstein, 24105 Kiel, Germany; n.margraf@neurologie.uni-kiel.de; 2Institute of Clinical Chemistry, Christian-Albrecht University of Kiel and University Medical Center Schleswig-Holstein, 24105 Kiel, Germany; justina.dargvainiene@uksh.de

**Keywords:** polyspike, aseptic meningoencephalitis, electroencephalogram

## Abstract

We report the electroencephalography (EEG) showing an intermittent generalized polyspike pattern in EEG due to an aseptic meningoencephalitis in a 71-year-old soporous patient. Initially, she presented with word-finding disturbances and later with generalized tonic–clonic seizures. The cerebrospinal fluid (CSF) showed pleocytosis of 99 leukocytes/μL (primarily neutrophils) and an increased protein level of 1240 mg/L (CSF/serum glucose ratio and lactate unremarkable). Pathogens and autoimmune antibodies in CSF were not found. Brain imaging was unremarkable. After antibiotic, antiviral and anticonvulsive therapy, the pattern in the EEG was no longer detectable. The patient was discharged to go home due to absence of any residues.

**Figure 1 diagnostics-13-02569-f001:**
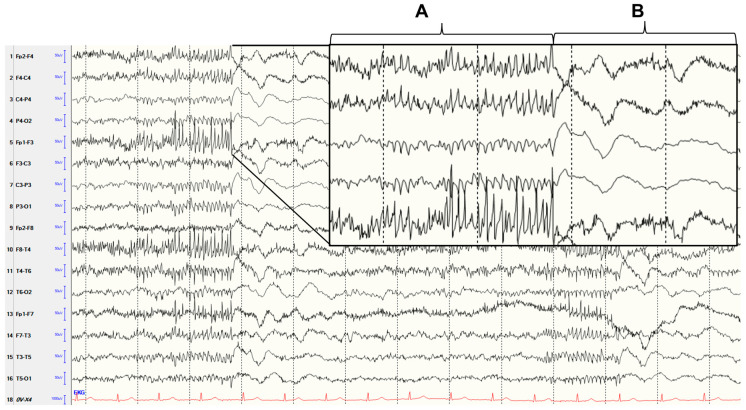
This electroencephalography (EEG) (10–20 system, sequential montage, sensitivity: 10 µV/mm, high frequency filters: 70 Hz, time base: 15 mm/sec) showing an intermittent generalized polyspike pattern was found in a 71-year-old soporous patient due to an aseptic meningoencephalitis. Polyspikes (**A**) were followed by delta waves (**B**). Initially, the patient presented with word-finding disturbances and later with generalized tonic–clonic seizures. The cerebrospinal fluid (CSF) showed pleocytosis of 99 leukocytes/μL (primarily neutrophils) and an increased protein level of 1240 mg/L (CSF/serum glucose ratio and lactate unremarkable, identical oligoclonal bands in serum and liquor). Pathogens (Escherichia coli, Haemophilus influenzae, Listeria monocytogenes, Neisseria meningitidis, Streptococcus agalactiae, Streptococcus pneumoniae, Cytomegalovirus, Enterovirus, Herpes simplex virus 1, Herpes simplex virus 2, Varicella zoster virus, Cytomegalovirus, Human herpesvirus 6, Human parechovirus, Cryptococcus neoformans/gattii, Tick Borne Encephalitis virus and Borrelia burgdorferi) and autoimmune antibodies (antibodies against amphiphysin, contactin-associated protein-2 [CASPR 2], collapsin response mediator protein 5 [CRMP-5], gamma-aminobutyric acid-b [GABA b] receptor, leucine-rich glioma inactivated 1 [LGI 1], Ma-proteins, N-methyl-D-aspartate [NMDA] receptor, dipeptidyl-peptidase-like protein 6 [DPPX], alpha-amino-3-hydroxy-5-methyl-4-isoxazolepropionic acid [AMPA]-receptor, Hu, Yo, Ri, glutamic acid decarboxylase [GAD] and myelin oligodendrocyte glycoprotein [MOG]) were not found in CSF. Magnetic resonance imaging (MRI) of the brain with contrast agent was unremarkable. Therapy was conducted with levetiracetam (2 × 1 g/die), lacosamide (2 × 100 mg/die), ceftriaxone, ampicillin and acyclovir. After clinical improvement, the pattern in the EEG was no longer detectable. The CSF after 8 days was normalized (cell count 3 leukocytes/μL). The patient was discharged to go home due to absence of any residues. Ten months later, the patient was examined in the outpatient clinic. She reported that no seizures occurred. The EEG was normal with no epileptiform discharges. A polyspike pattern in EEG represents interictal epileptiform discharges. It has been found in different immunological and structural conditions. Among these are encephalitis with MOG-antibodies [1] and with anti-ganglioside antibodies [2] or Hashimoto encephalopathy [3]. In general, EEG changes were found in patients with inflammatory processes in the brain [4]. Slow waves, epileptiform discharges and electroencephalic seizures were common in herpes encephalitis [5]. In autoimmune encephalitis, generalized or focal slowing, epileptiform discharges and electroencephalographic seizures were found [6]. Up to 28% of patients with aseptic meningitis without seizures had generalized and focal slowing [7]. In the reported case, no pathogens or autoimmune antibodies were found. Brain MRI also showed no structural changes. However, considering the symptoms with reported and observed generalized tonic-clonic seizures [8], the clinical course with improvement after treatment initiation, and the absence of the polyspike pattern on EEG after treatment initiation, it is most likely that the polyspike pattern was associated with meningoencephalitis. Although the cause of the meningoencephalitis (i.e., aseptic) was not found in the case reported here, neurologists should be aware that a polyspike pattern in EEG may be associated with an inflammatory process in the brain. Therefore, appropriate diagnostics should be performed if a polyspike pattern is found in the EEG.

## Data Availability

Not applicable.

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
