# Peer review of "Generalized Polyspike Pattern in EEG Due to Aseptic Meningoencephalitis"

_diagnostics, 2023, doi:10.3390/diagnostics13152569_

Round 1

Reviewer 1 Report

The authors try to report an interesting case.

However, there are some concerns about this case report.

First, the diagnosis of this patient was not clear. It was a case of the aseptic meningoencephalitis, but the exact cause is unknown. Second, it was not clear whether the generalized polyspike observed in this patient was really related to meningoencephalitis or not. Is there any possibility that this is an EEG finding that was present before the onset of this disease?

Author Response

We thank the reviewer for the time spent in reading and commenting our manuscript. We concerns of the reviewer are well-founded. We hope to address the concerns well.

Reviewer 1:

Comment 1: First, the diagnosis of this patient was not clear. It was a case of the aseptic meningoencephalitis, but the exact cause is unknown.

Answer: This is correct. Unfortunately, the exact reason of the meningoencephalitis is unclear. We did neither find a pathogen nor an (autoimmune) antibody. In order to clarify this point, we added the following sentences to the manuscript:

"In the reported case, no pathogens or autoimmune antibodies were found. Brain MRI also showed no structural changes."

Comment 2: Second, it was not clear whether the generalized polyspike observed in this patient was really related to meningoencephalitis or not. Is there any possibility that this is an EEG finding that was present before the onset of this disease?

Answer: We thank the review for this comment. This is the most important question. We think it is unlikely that this EEG finding is independent of the reported disease due to the following reasons:

This pattern was found in the first EEG of the patient we did. We have additional three EEGs of the same stay after begin of the treatment and after substantial clinical improvement. These three EEGs did not show a polyspike pattern. One EEG 10 months later did not show a polyspike pattern, too.

This pattern is a pattern of interictal epileptiform discharges reported in patients with epileptic seizures. We observe epileptic seizures during the stay and one seizure was reported before the arrivial in our emergency department. Afterwards the patients did not have further epileptic seizures.

As discussed in the manuscript polyspike patters were reported in inflammatory processes of the brain.

We added the following sentencse to the manuscript:

"Ten months later, the patient was seen in the outpatient clinic. She reported that no seizures occurred. The EEG was normal with no epileptiform discharges."

and

"However, considering the symptoms with reported and observed seizures, the clinical course with improvement after treatment initiation, and the absence of the polyspike pattern on EEG after treatment initiation, it is most likely that the polyspike pattern was associated with meningoencephalitis."

Reviewer 2 Report

In the manuscript entitled “Generalized polyspike pattern on EEG due to aseptic meningoencephalitis”, Markus A. Hobert and colleagues reported the intermittent generalized polyspike pattern EEG from a case of aseptic meningoencephalitis. This is an interesting image report. However, there are many issues that the authors need to address.

1.The EEG is the abbreviation of electroencephalography. Electroencephalography should be first shown in the manuscript.

2. Actually, the EEG is always abnormal in the viral encephalitis. More citations and discussion are necessary. (ex. Headache. 2001;41:79-83.)

3.  Is there any special in the brain images of the case?  Please try to combine relevant diagnostic data to show the unique and interesting of the case.

I believe that after revisions, this research paper may have good quality.

In the manuscript entitled “Generalized polyspike pattern on EEG due to aseptic meningoencephalitis”, Markus A. Hobert and colleagues reported the intermittent generalized polyspike pattern EEG from a case of aseptic meningoencephalitis. This is an interesting image report. However, there are many issues that the authors need to address.

1.The EEG is the abbreviation of electroencephalography. Electroencephalography should be first shown in the manuscript.

2. Actually, the EEG is always abnormal in the viral encephalitis. More citations and discussion are necessary. (ex. Headache. 2001;41:79-83.)

3.  Is there any special in the brain images of the case?  Please try to combine relevant diagnostic data to show the unique and interesting of the case.

I believe that after revisions, this research paper may have good quality.

Author Response

We thank the reviewer for the time and effort reading and commenting our manuscript. We think the comments are very useful and help us to improve the manuscript.

Comment 1: The EEG is the abbreviation of electroencephalography. Electroencephalography should be first shown in the manuscript.

Answer: We followed the suggestion and added Electroencephalography (EEG) in the abstract and the description of the figure, when shown first.

Comment 2: Actually, the EEG is always abnormal in the viral encephalitis. More citations and discussion are necessary. (ex. Headache. 2001;41:79-83.)

Answer: We thank the reviewer for this comment and literature recommendation. We added the reference and extended the discussion. It reads now as follows:

"A polyspike pattern in EEG represents interictal epileptiform discharges. It has been found in different immunological and structural conditions. Among these are e.g. the encephalitis with MOG-antibodies [1] and with anti-ganglioside antibodies (IgG-GQ1b, GD1a, and GT1b)[2] or a hashimoto encephalopathy[3]. In general, EEG changes were found in patients with inflammatory processes in the brain [4]. Slow waves, epileptiform discharges, electroencephalic seizures and were common in herpes encephalitis[5]. In autimmune encephalitis generalized or fokal slowing, epileptiform discharges, electroencephalographic seizures were found [6]. Up to 28% of patients with aseptic meningitis without seizures had generalized and focal slowing [7].

In the reported case, no pathogens or autoimmune antibodies were found. Brain MRI also showed no structural changes. However, considering the symptoms with reported and observed seizures, the clinical course with improvement after treatment initiation, and the absence of the polyspike pattern on EEG after treatment initiation, it is most likely that the polyspike pattern was associated with meningoencephalitis."

Comment 3:  Is there any special in the brain images of the case?  Please try to combine relevant diagnostic data to show the unique and interesting of the case.

Answer: There were no abnormalities in the MRI of the brain.

We also thank the reviewer for this comment. The interesting aspect in this case is the rare EEG pattern associated with meningoencephalitis. We have tried to clarify this aspect in the manuscript. It now reads as follows:

"In the reported case, no pathogens or autoimmune antibodies were found. Brain MRI also showed no structural changes. However, considering the symptoms with reported and observed seizures, the clinical course with improvement after treatment initiation, and the absence of the polyspike pattern on EEG after treatment initiation, it is most likely that the polyspike pattern was associated with meningoencephalitis."

And, as our main message was, that this EEG pattern should trigger diagnostics we added the following sentence:

"Therefore, appropriate diagnostics should be performed if a polyspike pattern is found in the EEG."

Round 2

Reviewer 1 Report

You have worked hard to improve your manuscript, but it is not enough.

Author Response

Thank you again for reviewing our manuscript.

To further improve the manuscript, specific recommendations and ideas would be very helpful for us.

Regarding the aspects of round 1 review, we feel that we have sufficiently addressed comment 1. The course of aseptic meningoencephalitis is unknown. As the meningoencephalitis is aseptic, by definition no pathogen was found. Autoimmune antibodies were also negative. As we are reporting an interesting image (i.e. the focus is the figure), this unresolved diagnostic aspect seems acceptable to us in this type of publication.

The concern in Comment 2 has been addressed by adding the following paragraph:

In the reported case, no pathogens or autoimmune antibodies were found. Brain MRI also showed no structural changes. However, considering the symptoms with reported and observed generalized tonic-clonic seizures, the clinical course with improvement after treatment initiation, and the absence of the polyspike pattern on EEG after treatment initiation, it is most likely that the polyspike pattern was associated with meningoencephalitis.

Reviewer 2 Report

In the manuscript entitled “Generalized polyspike pattern on EEG due to aseptic meningoencephalitis”, Markus A. Hobert and colleagues had revised and corrected in the revision. However, there are some issues that the authors need to address.

1.   There are also many abbreviations including CSF, CMRT, CASPR 2, CV2/CRMP5, GABA b etc. in the manuscript. Please carefully check and correct.

2.   Which seizure type was in the case by 2017 ILAE Classification? Please extend in the discussion with citation.

I believe that after minor revisions, this research paper may have good quality.

Author Response

We thank the reviewer again for the time and effort spent in reading and commenting on our manuscript. We hope that we have addressed the reviewer's comments.

Comment 1: There are also many abbreviations including CSF, CMRT, CASPR 2, CV2/CRMP5, GABA b etc. in the manuscript. Please carefully check and correct.

Response 1: We thank the reviewer for this note and have used the spelled out terms of the abbreviations

Comment 2: Which seizure type was in the case by 2017 ILAE Classification? Please extend in the discussion with citation.

Response 2: We thank the reviewer also for this aspect and have added "generalized tonic-clonic [seizures] (Fischer RS et al. 2017)".